# Granulovirus PK-1 kinase activity relies on a side-to-side dimerization mode centered on the regulatory αC helix

Michael R. Oliver[1,6,8], Christopher R. Horne [2,3,8], Safal Shrestha[4], Jeremy R. Keown [1,7], Lung-Yu Liang[2,3], Samuel N. Young[2], Jarrod J. Sandow [2,3], Andrew I. Webb [2,3], David C. Goldstone [1], Isabelle S. Lucet [2,3], Natarajan Kannan[4,5], Peter Metcalf [1✉] & James M. Murphy [2,3✉]

The life cycle of *Baculoviridae* family insect viruses depends on the viral protein kinase, PK-1, to phosphorylate the regulatory protein, p6.9, to induce baculoviral genome release. Here, we report the crystal structure of *Cydia pomenella* granulovirus PK-1, which, owing to its likely ancestral origin among host cell AGC kinases, exhibits a eukaryotic protein kinase fold. PK-1 occurs as a rigid dimer, where an antiparallel arrangement of the αC helices at the dimer core stabilizes PK-1 in a closed, active conformation. Dimerization is facilitated by C-lobe:C-lobe and N-lobe:N-lobe interactions between protomers, including the domain-swapping of an N-terminal helix that crowns a contiguous β-sheet formed by the two N-lobes. PK-1 retains a dimeric conformation in solution, which is crucial for catalytic activity. Our studies raise the prospect that parallel, side-to-side dimeric arrangements that lock kinase domains in a catalytically-active conformation could function more broadly as a regulatory mechanism among eukaryotic protein kinases.

---

[1] School of Biological Sciences, University of Auckland, Auckland, New Zealand. [2] Walter and Eliza Hall Institute of Medical Research, Parkville, VIC, Australia. [3] Department of Medical Biology, University of Melbourne, Parkville, VIC, Australia. [4] Institute of Bioinformatics, University of Georgia, Athens, GA, USA. [5] Department of Biochemistry and Molecular Biology, University of Georgia, Athens, GA, USA. [6] Present address: School of Biological Sciences, University of Edinburgh, Edinburgh, UK. [7] Present address: Division of Structural Biology, Wellcome Centre for Human Genetics, University of Oxford, Oxford, UK. [8] These authors contributed equally: Michael R. Oliver, Christopher R. Horne. ✉email: peter.metcalf@auckland.ac.nz; jamesm@wehi.edu.au

*B*aculoviridae are a widespread family of large DNA viruses that specifically infect the orders of *Lepidoptera*, *Hymenoptera* and *Diptera* holometabolic insects[1,2]. Following primary infection and localization to the host cell nucleus, the initiation of baculoviral gene transcription, DNA replication and nucleocapsid assembly occurs within an electron-dense structure called the virogenic stroma[2,3]. A key modulator of this process is a highly-conserved, small, positively-charged DNA-binding protein named p6.9, which is expressed by all members of the *Baculoviridae* family[4,5]. p6.9 shares structural similarity to the protamines that are synthesised late in spermiogenesis to replace histones and condense the spermatid genome into a genetically inactive state[6,7]. These similarities include an arginine-rich sequence that confers the ability to bind DNA, neutralize the negative charge of the phosphate backbone and condense DNA into a highly compact chromatin-like structure[8]. In baculoviruses, this p6.9-mediated condensation is required for the encapsulation of the viral genome inside the nucleocapsid core[9]. However, much like protamine, the p6.9–DNA interaction can be attenuated by post-translational phosphorylation, leading to the release of baculoviral DNA from the nucleocapsid, triggering a cascade of early to very-late gene transcription[10–13]. Therefore, both the encapsulation and release of the baculoviral genome is regulated by the phosphorylation status of p6.9, thus supporting a pivotal role for post-translational modifications in the baculovirus life cycle.

Phosphorylation of p6.9 is mediated by protein kinase-1 (PK-1), a protein kinase expressed within the nucleocapsid of lepidopteran-infecting baculoviruses[14–17]. Deletion of PK-1 compromises encapsulation of the baculoviral genome in nucleocapsid assembly[16], and mutation of 7 PK-1 substrate Ser/Thr residues in p6.9 markedly reduced the hyper-expression of very-late genes[17], consistent with PK-1 serving a crucial role in the regulation of viral proliferation. PK-1 primarily phosphorylates the N-terminal region of p6.9 and has been reported to promote phosphorylation of arginine residues present within this region[11,17], although the precise molecular mechanism is currently unclear.

Here, we present the crystal structure of *Cydia pomenella* granulovirus PK-1: a serine/threonine protein kinase that, despite being encoded by the baculoviral genome, shares the fold of a eukaryotic protein kinase. Phylogenetically, PK-1 appears to have originated from its host's genome, with the greatest similarities to the AGC kinase domain of ribosomal protein S6 kinase A5 (RPS6KA5) homologues in arthropods. The acquisition of an atypical N-terminal helix, which is peculiar to PK-1 orthologs in granuloviruses and some nucleopolyhedroviruses, facilitates constitutive dimerization via domain swapping between protomers, and is augmented by a contiguous β-sheet between the β1–β5 strands of each protomer, and interactions between the αC helices via an antiparallel arrangement. Biochemical studies support the idea that this dimerization mode locks the PK-1 dimer into a catalytically-active conformation and that dimerization is crucial to its catalytic activity. PK-1 harbours ATP-binding and catalytic residues typical of classical eukaryotic protein kinases within its active site, although in contrast to conventional protein kinases, the locked active conformation reduces the dependence on the N-lobe β3-strand Lys and the Gly-rich loop for ATP-binding. The unusual parallel, side-to-side arrangement of kinase domains within the dimer, characterized by the N-lobe:N-lobe and C-lobe:C-lobe interactions revealed by our crystal structure, positions each active site facing in opposing directions and primed for substrate engagement. These findings suggest that stabilizing a catalytically-active conformation via a parallel side-to-side dimerization mode could represent a mechanism used by other protein kinases to regulate their catalytic activities.

## Results

### *C. pomenella* GV PK-1 adopts an active kinase conformation.

To gain atomic level insight into the PK-1 structure and an understanding of the regulation of its catalytic activity, we solved the crystal structure of PK-1 from *C. pomenella* granulovirus (*Cp*GV) in complex with AMP at 2 Å resolution (Fig. 1; coding sequence in Supplementary Table 1; crystallographic statistics in Supplementary Table 2). PK-1 exhibited a bi-lobal architecture, comprising a smaller N-terminal lobe (Fig. 1a, tan) and a large C-terminal lobe (Fig. 1a, green) that are connected by a short hinge region (Fig. 1a, purple), which resembles the classical eukaryotic protein kinase fold[18]. The N-lobe contains five β-strands (β1–β5) that form an antiparallel β-sheet, the regulatory αC-helix (Fig. 1a, dark blue) and an unusual additional α-helix at the very N-terminus that sits atop the N-lobe. The C-lobe is typical of protein kinases, and mostly comprises α-helices (αD–αI), in addition to the activation loop (Fig. 1a, orange) and two additional β-strands (β7–β8).

The crystal structure contained two PK-1 protomers in the asymmetric unit, which closely associate to form a parallel, side-to-side dimeric assembly (Fig. 1b, Supplementary Fig. 2a). Analysis using the PDBePISA server[19] confirmed that the dimer interface was extensive, with a buried surface area of ~1652 Å$^2$ and ΔG of −20.6 kcal/mol, consistent with stable association. This dimerization interface is formed by four key regions, which include: an atypical N-terminal α-helix that is domain-swapped between protomers (Supplementary Fig. 2b); a contiguous β-sheet formed by the N-lobe β1–β5 strands of each protomer (Supplementary Fig. 2c); a hydrophobic core stabilized by aromatic stacking interactions between the N-lobe β-sheet and the αC-helix, primarily through Phe61 and Phe79 (Supplementary Fig. 2c); and a network of van der Waals interactions and hydrogen bonds between the activation loop and the C-terminus of the αE-helix from opposing protomers (Supplementary Fig. 2d). As a consequence of these interactions, the αC helices from each protomer are positioned side-by-side in an antiparallel arrangement (Fig. 1b) and the active sites of each protomer face in opposite directions. Mapping the electrostatic potential to the molecular surface of PK-1 revealed a cluster of negative charge within the αF–αG loop in the C-lobe, near the active site (Fig. 1c).

The active site of PK-1 grossly resembles that of a canonical eukaryotic protein kinase, and the core catalytic residues[20]—the N-lobe β3-strand Lys, the catalytic loop Asp and activation loop Asp—are all present and conserved among PK-1 orthologs (Fig. 1d). Accordingly, co-crystallized AMP was bound in the active site in a conformation analogous to that commonly observed for ATP in eukaryotic protein kinases (Fig. 1d). The closed conformation of the PK-1 kinase domain resembles the structures of active conformations of eukaryotic protein kinases[21,22], which appears to be stabilized allosterically by the αC helix of the dimeric partner PK-1 protomer (Fig. 1d, e). The αC helix from one PK-1 protomer associates with that of its dimer partner in an antiparallel fashion (Fig. 1e) anchored by hydrophobic interactions between Phe61 in the αC helix of one protomer and Phe56 of the dimer partner αC helix. This conformation is augmented by hydrophobic interactions of Phe61 (αC helix) with Phe79 and Phe81 (β4 strand, N-lobe), the latter of which also forms hydrophobic contacts with Ile65 (αC helix) (Supplementary Fig. 2c). In addition, hydrogen bonds between the Asp62 pairs and with Ser58 of the dimer partner contribute to the dimer interface between αC helices (Fig. 1e). In addition to the ATP-binding β3-strand residue, Lys49, forming a salt bridge with Glu60 of the αC-helix (the counterpart of the Lys72:Glu91 salt bridge in protein kinase A), we observed an intact regulatory (R) spine comprising His64, Leu76, His139 and Tyr161 (Fig. 1f). Although unusual in protein kinase structures,

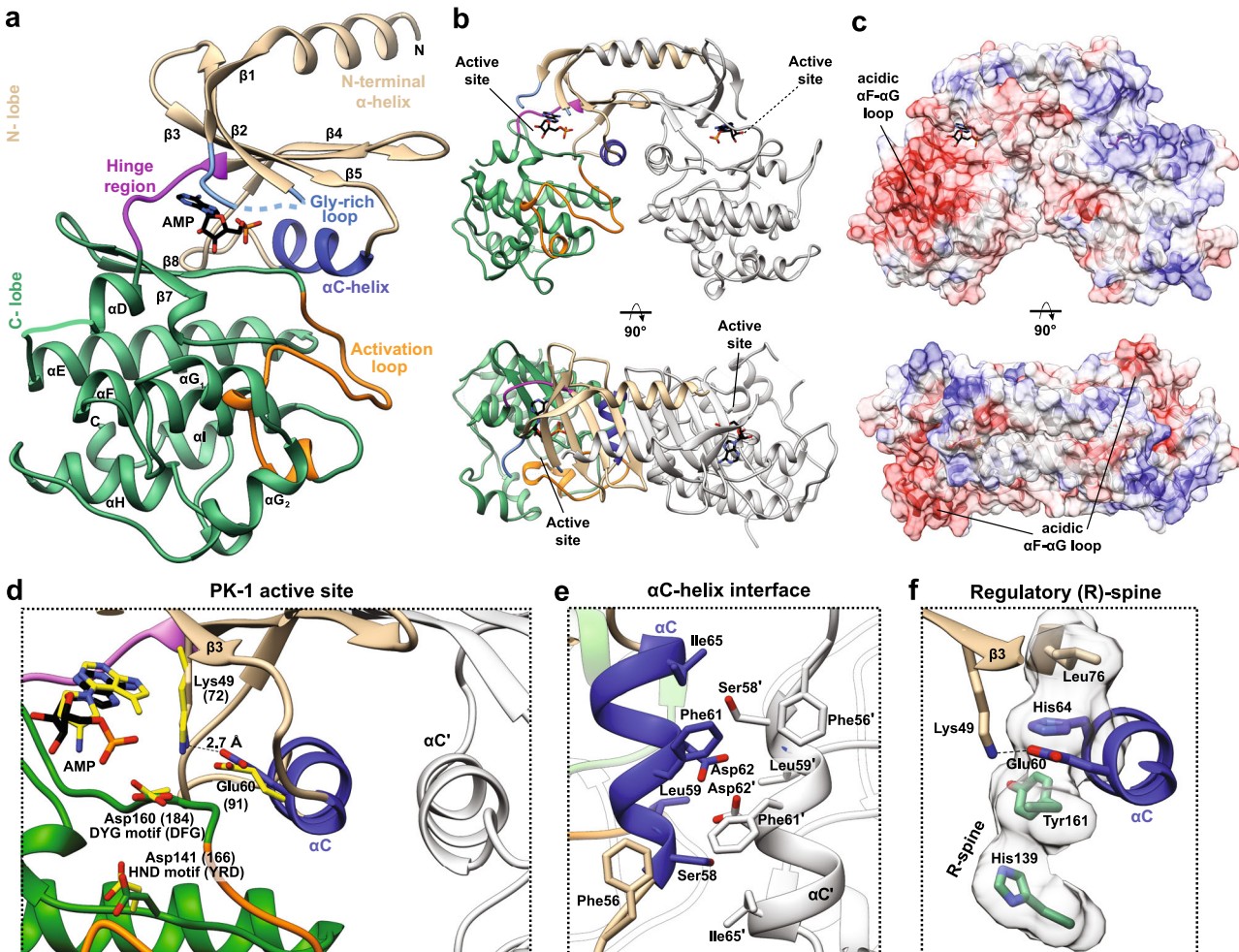

**Fig. 1 Overall structure of *Cp*GV PK-1. a** The PK-1 catalytic core has two lobes—an N-terminal lobe (tan) and a C-terminal lobe (green), connected via a short hinge region (purple). The N-terminal lobe contains an extended α-helix at the N-terminus, a bound AMP nucleotide (black), the Gly-rich loop (light blue, disordered) and the αC-helix (dark blue). The C-terminal lobe is largely α-helical and contains the activation loop (orange). **b** Cartoon representation of the side-to-side dimeric assembly formed by PK-1 (coloured by **a**, grey). The active site is located within a cleft formed between the bi-lobed architecture, which face in opposite directions between each protomer. **c** The electrostatic potential mapped onto the molecular surface of PK-1 shows a cluster of negative charge on the front face (red) at the αF-αG loop. **d** The active site residues of PK-1 (shown as sticks) grossly resemble those of the canonical eukaryotic protein kinase A (PKA; PDB ID: 3MVJ)[75] (yellow sticks, residue number and catalytic motifs in parentheses). Conventional protein kinases contain an ATP-binding VAI$\underline{K}$ motif in the β3-strand (present as FVCK in PK-1, YAMK in PKA), a $Mg^{2+}$ binding $\underline{DFG}$ motif at the start of the activation loop (DYG in PK-1, DFG in PKA) and the catalytic H$\underline{RD}$ motif in the catalytic loop (HND in PK-1, YRD in PKA). AMP is bound in a conformation analogous to that of the nucleotide in PKA (yellow). **e** In PK-1, an antiparallel arrangement between dimer-related αC helices, mediated principally by hydrophobic interactions, allosterically locks PK-1 in an activated conformation. **f** The residues that form an intact regulatory (R)-spine are shown as sticks and by surface representation. As a result, the salt bridge between the β3-strand Lys (Lys49) and αC-helix Glu (Glu60) is preserved.

the entire activation loop was resolved in our PK-1 structure. The activation loop adopted an extended conformation, which appears to have been stabilized by interactions with its dimeric partner, including residues C-terminal to the αE-helix located at the dimeric interface (Supplementary Fig. 2d, e).

**PK-1 sequences diverge from eukaryotic protein kinases.** Although encoded by a granulovirus genome, our structure revealed PK-1 to possess a eukaryotic protein kinase fold. This led us to propose the origin of the kinase might lie in acquisition from the host. We performed Basic Local Alignment Search Tool (BLAST)[23] searches to identify sequences closely related to *Cp*GV PK-1, followed by phylogenetic analysis to determine the relationships between PK-1 and other eukaryotic protein kinases. We identified PK-1 homologues in granuloviruses (98–100% coverage, and 44-100% identity) and nucleopolyhedroviruses (61–99% coverage and 33–46% sequence identity). In several granulovirus

sequences and a subset of nucleopolyhedrovirus sequences, sequence homology extended beyond the kinase domain to include the N-terminal extension that formed a domain-swapped helix in the *Cp*GV PK-1 structure. However, in most distantly related viral and eukaryotic protein kinase sequences, sequence similarity with PK-1 is limited to the catalytic domain, suggesting that the N-terminal helix is specific to the baculovirus clade, and likely acquired after incorporation into the viral genome. Remarkably, viral PK-1 sequences were found to be most closely related to metazoan homologues of RPS6KA5 kinases in the AGC family (bootstrap value of 0.874), and to a lesser extent to metazoan homologues of RPS6KA5 kinases belonging to $Ca^{2+}$/calmodulin protein kinase (CAMK) family (Fig. 2a). Because the greatest sequence similarity was to those from arthropods, such as *Tetranychus urticae* (Red spider mite), we propose that PK-1 was acquired from the genome of the insect cell host. Surprisingly, we identified an ortholog (WP_143445091.1) in

## a

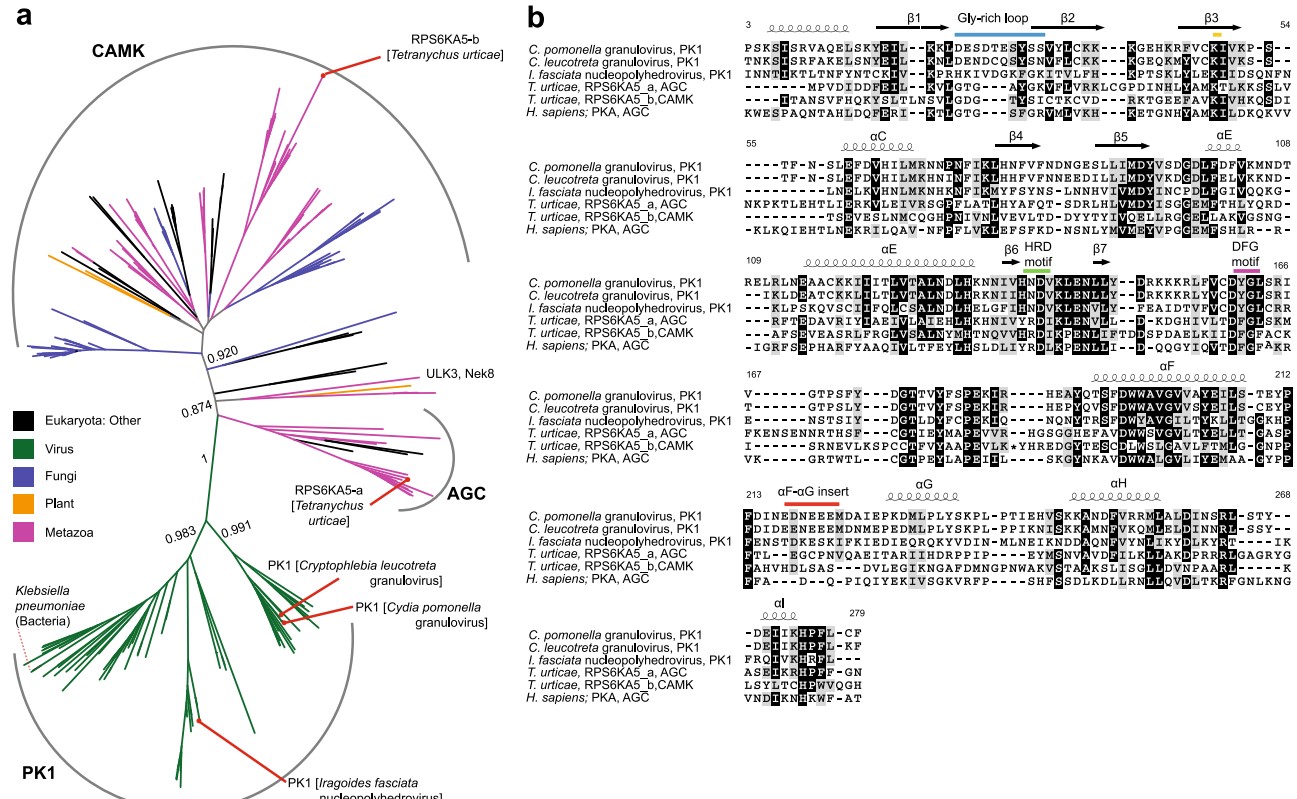

## b

**Fig. 2 Phylogenetic analysis of baculoviral PK-1. a** Unrooted tree of PK-1 and closely related sequences as identified through a BLAST search. Types of organisms are differentiated by colour; major bootstrap confidence levels are shown and each prominent kinase family are highlighted. **b** Multiple sequence alignment of CpGV PK-1 with other representative sequences on the phylogenetic tree in (**a**). The alignment was visualized using ESPript 3.0[76] with CpGV PK-1 crystal structure as input to define the secondary structures. The black background highlights identical residues, while the grey background indicates similar residues. Residues are labelled with respect to CpGV PK-1. The Gly-rich loop (blue), β3-strand Lys (orange), canonical HRD (green) and DFG (pink) motif, and acidic αF-αG insert loop (red) are highlighted. Secondary structure of CpGV PK-1 is shown above the alignment. Accession numbers for the sequence analysis are as follows: C. leucotreta PK-1: NP_891851.1; I. fasciata PK-1: ACJ04629.1; T. urticae AGC, Ca$^{2+}$/calmodulin-dependent protein kinase (CAMK): XP_015782813.1; H. sapiens PKA: NP_002721.1. * denotes an insert sequence in RPS6KA5.

Klebsiella pneumoniae, a bacterium, raising the prospect of horizontal gene transfer from baculoviruses to prokaryotes or convergent evolution following acquisition from an insect host.

Next, we sought to identify evolutionary constraints distinguishing viral PK-1 and related CpGV sequences from other eukaryotic protein kinase sequences (ePKs). To quantify the constraints, we performed a Bayesian statistical analysis on the sequence alignment containing PK-1 and ePK sequences. The analysis revealed constraints in several key regions, including the catalytic loop, the activation loop, and the αF-helix (Supplementary Fig. 3a, b). Specifically, two different clusters of interactions were formed by the PK-1 specific residue constraints in the N- and C-termini of the activation loop (Supplementary Fig. 3b, Insets). At the N-terminus of the activation loop, Y161 (the equivalent of the Phe in the DFG motif) forms van der Waals contacts with H64 and F73. Hydrophobic interactions are extended by van der Waals contacts between M67, N70 and F73 (Supplementary Fig. 3b, Top Inset). At the C-terminal end of the activation loop, a salt bridge between D174 in the activation loop and K184 immediately following the end of the activation loop is further stabilized by additional hydrogen bonds from N140 in the catalytic loop, with K184 further stabilized by van der Waal contacts with F180 and F194 (Supplementary Fig. 3b, Bottom Inset). Together, these PK-1-specific variations appear to further stabilize the activation loop in an extended conformation.

The CpGV PK-1 active site contains the β3-strand Lys, and Asp residues in the catalytic (HRD motif; HND in PK-1) and

activation (DFG motif; DYG in PK-1) loops, which are typical motifs associated with protein kinase activity, as mentioned above. We note that despite the substitution of Phe in the canonical DFG motif, the Tyr present still contributes to the assembly of an intact R-spine in our CpGV crystal structure (Fig. 1d). Two structural elements, however, are of interest owing to their divergence relative to conventional protein kinases: a loss of Gly residues from the Gly-rich loop; and an acidic insert in the loop connecting the αF and αG helices (Fig. 2b). Located between the β1- and β2-strands of the N-lobe, the Gly-rich loop is a highly conserved motif among kinase proteins[24–26]. The component GδGXϕG (G, Gly; δ, hydrophilic; X, any amino acid; ϕ, large hydrophobic amino acid) consensus motif facilitates loop mobility to shield the ATP substrate from bulk solvent and position the ATP γ-phosphate for catalysis[27–29]. Unusually, the corresponding sequence in CpGV PK-1, DESdteSYS (where dte is an insert sequence), and those of other granulovirus PK-1 orthologs (Fig. 2b; Supplementary Fig. 3), lack all three conserved glycine residues, while only one of the glycine residues is lost in nucleopolyhedrovirus homologues (Fig. 2b; Supplementary Fig. 3). This divergence in the granulovirus PK-1 Gly-rich loop is reminiscent of that observed among small molecule kinases, where glycine residues are commonly lost from this loop. In contrast to typical protein kinases, the loss of Gly-rich loop conformational flexibility is believed to confer an evolutionary advantage upon small molecule kinases by promoting efficient, constitutive substrate

phosphorylation[30,31], raising the prospect of a similar function in viral protein kinases.

Another atypical feature of PK-1 is an insertion of acidic residues in the αF-αG loop in the C-lobe (Fig. 2b; Supplementary Fig. 3), which are surface-exposed (Fig. 1c). This loop is known to contribute to substrate recognition in other kinases[32,33], suggesting that the sequence, E$^{217}$DNEEE$^{222}$, might contribute to recognition of highly-basic substrates, such as p6.9. Consistent with such a conserved function, an acidic sequence in this loop is conserved among granulovirus PK-1 orthologs, but not within nucleopolyhedrovirus orthologs (Fig. 2b). Furthermore, in the crystal structure, the acidic insert lines a putative substrate binding pocket, presumably for selectively binding residues in the p6.9 substrate.

**PK-1 adopts a rigid dimeric conformation in solution.** Because PK-1 occurred as an intertwined dimer within our crystal structure, we sought to evaluate its oligomeric state in solution by performing sedimentation velocity analytical ultracentrifugation (AUC) experiments at a concentration of 0.9 mg mL$^{-1}$ (27 μM). When the sedimentation data were fitted with a continuous sedimentation coefficient [c(s)] distribution model, a single species was observed with a sedimentation coefficient of 4.4S and a frictional ratio ($f/f_0$) of 1.29, suggesting that PK-1 is asymmetric (Fig. 3a; statistics shown in Supplementary Table 3). Furthermore, the measured molar mass for this single species was 66.8 kDa, consistent with a calculated dimer mass of 66 kDa, confirming that PK-1 exists as a dimer in solution.

We then proceeded to characterise the arrangement of the dimer in solution using small-angle X-ray scattering (SAXS)

experiments with an inline size exclusion chromatography (SEC) setup (statistics shown in Supplementary Table 4). As illustrated in the inset accompanying the experimental scattering profile (Fig. 3b), the Guinier analysis produced a linear plot, consistent with a single monodisperse species in solution and an absence of substantive aggregation or interparticle interference. The theoretical scattering profile, which was generated from the $Cp$GV PK-1 crystal structure was consistent with the experimental data ($\chi^2$ value of 0.39 using CRYSOL; Fig. 3b), which supports that the arrangement of protomers is biologically relevant in solution and not just a feature within the crystal. The maximum particle dimension ($D_{max}$) of PK-1 was determined to be 96 Å from the $P(r)$ analysis (Fig. 3c), which is in agreement with the envelope, based on the crystal structure, calculated using CRYSOL (92 Å). The modest difference is likely associated with flexibility of the protomers in solution, an observation supported by the $f/f_0$ in our AUC experiments. Measured on a relative scale using SAXS-MoW2[34], the molecular weight of PK-1 from the SAXS experiment was 74.3 kDa, consistent with the calculated dimeric mass (66 kDa).

To further explore the molecular basis of this dimeric interface and assess overall flexibility, we subjected the apo form of the $Cp$GV PK-1 dimer to 1 microsecond (μs) long unbiased molecular dynamics (MD) simulations. During the simulation, the two chains in the dimeric form moved away from each other initially but remained bound throughout the simulation (Supplementary Movie 1). The movement away from each other is seen as an initial increase in the radius of gyration ($Rg$) followed by fluctuations around a mean $Rg$ of 2.8 nm (Supplementary Fig. 4). Interestingly, three regions distal to the dimer interface were dynamic, as indicated by their relatively higher root mean square

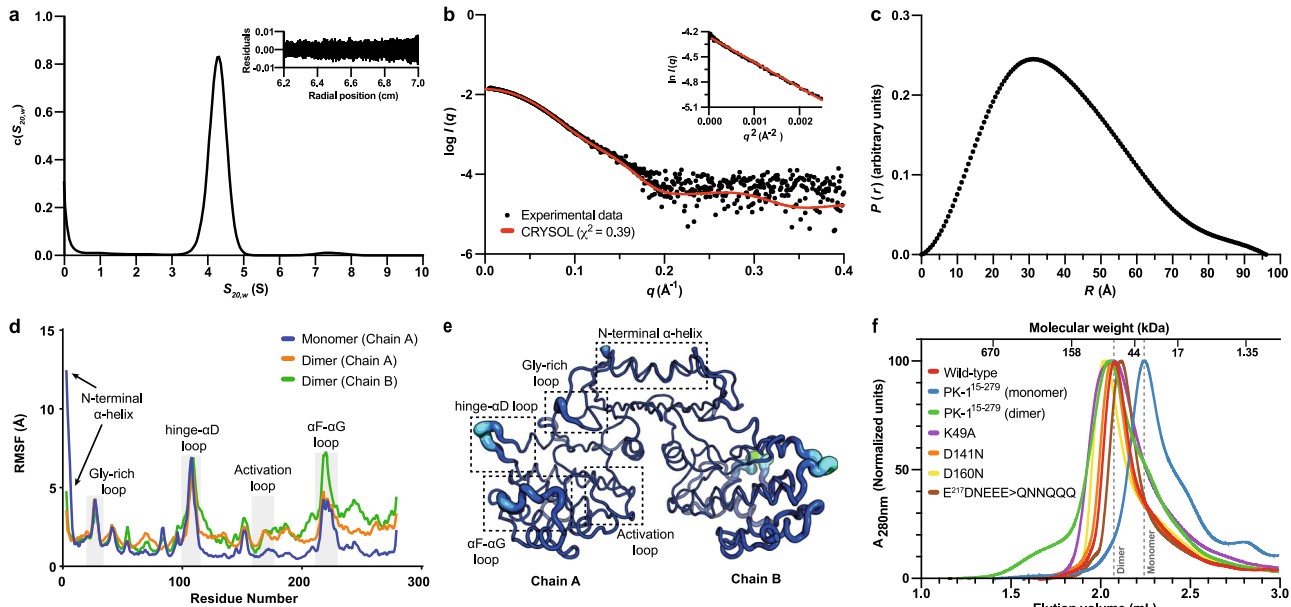

**Fig. 3 PK-1 adopts a rigid dimer assembly in solution. a** Sedimentation velocity data at a protein concentration of 0.9 mg mL$^{-1}$ were fitted to a continuous sedimentation coefficient [c(s)] distribution model. The residuals and best fit of the data are shown as an inset. **b** SAXS scatter profile for PK-1. The scattering data is best fit (red line, $\chi^2 = 0.39$) to the theoretical scatter of the PK-1 crystal structure. The Guinier plot is linear (inset), showing the sample is free from any measurable amounts of aggregation or inter-particle interference. **c** The pairwise distribution plot, calculated from the scattering data, estimate the maximum interparticle dimension ($D_{max}$) is 96 Å. **d** Plot showing the root mean square fluctuations (RMSF) of monomeric PK-1 (blue) and chain A and B of dimeric PK-1 (orange and green, respectively) in MD simulations. Residue fluctuations, monitored for the Cα atoms of the protein over the entire trajectory, are shown. The activation loop and most dynamic regions of PK-1 are highlighted. **e** Modelled from the MD simulation, the PK-1 dimer is represented as a putty cartoon, highlighting the atomic fluctuations of PK-1. These fluctuations are also graded by colour from lowest (dark blue) to highest (light blue) RMSF value. As per the RMSF plot in (**d**), the activation loop and most dynamic regions of PK-1 are shown. **f** Analytical SEC of wild-type PK-1 and mutant constructs. The elution volumes of molecular weight standards are shown above the chromatograms. Absorbance at 280 nm (A$_{280nm}$) is normalized across all samples.

fluctuations (RMSF) values (Fig. 3d, orange and green lines). These regions included the Gly-rich loop, hinge-αD loop and αF-αG loop (Fig. 3d, e). When the apo monomeric form of PK-1 was subjected to an approximately 2.5 µs long unbiased MD simulation, the same regions were also dynamic in addition to the N-terminal α-helix, which displayed higher RMSF values compared to the dimer (Fig. 3d, Supplementary Movie 2). Notably, in both simulations, the activation loop was relatively stable as indicated by low RMSF values. On mapping the PK-1 specific constraints to the monomeric MD trajectory, we find that the two clusters of interactions formed by PK-1 specific constraint residues at the N- and C-termini of the activation loop are stable (Supplementary Movie 2). This observation suggests that specific evolutionary changes within PK-1 might contribute to the stabilization of the activation loop in an extended conformation. Our observations further support the view that the activation loop – an element typically flexible in protein kinases – is rigidified in PK-1, presumably to facilitate constitutive substrate recognition and phosphorylation.

Based on this notion and the dimer observed in our crystal structure, we sought to investigate if a monomeric form of PK-1 could be obtained by generating a truncated PK-1 construct (PK-1$^{15-279}$) in which the atypical domain-swapped N-terminal helix is absent. While full-length wild-type PK-1 eluted as a single peak from SEC during purification, PK-1$^{15-279}$ eluted as two distinct peaks corresponding to dimeric (~66 kDa) and monomeric (~33 kDa) species (Fig. 3f), when analysed individually against a molecular weight standard, indicating this truncation of the N-terminal helix alters the oligomeric state. By comparison, substitutions of putative catalytic residues (K49A, D141N and D160N) and the acidic sequence in the αF-αG loop (E$^{217}$DNEEE>QNNQQQ) did not impact the oligomeric state, because these proteins eluted solely as dimeric species using analytical SEC. Taken together, our AUC, SAXS and SEC data, and molecular dynamics simulations demonstrate that full-length PK-1 exists as a rigid dimer in solution, which can be disrupted by truncation of the atypical, domain-swapped N-terminal helix.

**Biochemical characterization of PK-1**. We next sought to define the role of dimerization in regulating PK-1 catalytic activity by examining the activities of wild-type PK-1 relative to monomeric and dimeric PK-1$^{15-279}$, acidic αF–αG loop and active site mutant PK-1 proteins (Fig. 4a) in an ADP-Glo assay in the presence of a p6.9 substrate peptide. While wild-type PK-1 exhibited robust catalytic activity (Fig. 4b, red), the activity of the monomeric form of PK-1$^{15-279}$ was severely compromised (Fig. 4b, blue), with activity of ~20% of that of full-length wild-type PK-1. These data clearly implicate dimerization as serving a crucial function in promoting PK-1 catalytic activity. The dimeric form of the PK-1$^{15-279}$ truncation mutant retained ~70% of full-length wild-type PK-1 activity (Fig. 4b, green), suggesting that the active conformation dictated by the dimer, rather than dimerization per se, is crucial for optimal p6.9 phosphorylation.

PK-1 contains a conserved, atypical acidic sequence in αF-αG loop (E$^{217}$DNEEE) (Fig. 4a) that could potentially contribute to recognition of highly basic substrates, such as p6.9. We substituted this sequence to eliminate the negative charge (E$^{217}$DNEEE>QNNQQQ) and examined the impact on catalysis using the ADP-Glo assay. Surprisingly, this mutant exhibited comparable catalytic activity to full-length wild-type PK-1 (Fig. 4b, brown), indicating that in an in vitro context in which substrate is in excess, this sequence does not measurably enhance substrate recognition.

Because PK-1 adopts the classical eukaryotic protein kinase fold, it was of interest to establish whether the residues corresponding to core catalytic residues in conventional protein kinases serve comparable functions in PK-1. Accordingly, we compared the catalytic activities of K49A, D141N and D160N full-length PK-1 in ADP-Glo assays (Fig. 4a). As expected, the activities of the D141N and D160N PK-1 mutants were ablated (Fig. 4b, orange and yellow, respectively), illustrating that PK-1 catalytic activity relies on residues that are the counterparts of the catalytic residues within a conventional protein kinase. Unexpectedly, however, K49A PK-1 exhibited ~50% of wild-type PK-1 activity (Fig. 4b, purple). These data indicate that K49 in PK-1, which typically functions in positioning ATP via interaction with the α- and β-phosphates in conventional eukaryotic protein kinases, is dispensable for PK-1 catalytic activity. This raises the possibility that the active form of PK-1, stabilized by dimerization, and the Gly-rich loop sequence that is divergent from those of conventional protein kinases (Fig. 2a), obviates the reliance on K49 for ATP-binding and positioning for catalysis. To ensure the differences in activity we observed for each PK-1 construct were not attributable to compromised protein folding, we evaluated each variant's thermal stability by differential scanning fluorimetry. Wild-type and mutant PK-1 proteins exhibited melting temperatures ($T_m$) ranging from 50 to 61 °C (Fig. 4c), which are values typical of folded kinase domains[35]. Collectively, these data demonstrate that PK-1 is catalytically-active and that the assembly and conformation of PK-1 dimers is critical for optimal activity.

## Discussion

Here, we report the structure of a baculoviral kinase, PK-1, which adopts a fold typical of eukaryotic protein kinases and harbours active site residues typical of conventional eukaryotic kinases. Our sequence analyses indicate a eukaryotic ancestral origin of PK-1, where the kinase was acquired via co-opting features from multiple eukaryotic kinases, in particular RPS6KA5 from the insect host over its evolutionary trajectory. In contrast to most eukaryotic protein kinases, PK-1 occurs as a constitutive dimer. Dimerization via an atypical domain swapped N-terminal helix and the formation of a contiguous β-sheet between the two protomer's N-lobes positions the two protomers in a parallel side-to-side arrangement with an antiparallel arrangement of the two αC helices at the dimer core, such that the active sites of the two monomers face in opposite directions. The parallel side-to-side dimer adopted by PK-1 is unusual, because prior structural studies have reported only other modes of αC-helix-focused allosteric regulation to date (Fig. 5). Various modes of protein kinase allosteric regulation by protein ligands, such as cyclin binding to cyclin-dependent protein kinases (CDKs)[36] or the binding of ligands to the PDK-1 interacting fragment (PIF) pocket of PDK1[37], and kinases/pseudokinases via homo- and heterodimerization have been reported. In common with the parallel side-to-side dimerization mode that we report here for PK-1, each of the previously-reported binding modes influence the position of the central regulatory element, the αC helix, to dictate whether the kinase/pseudokinase adopts a closed conformation synonymous with the active form of a conventional protein kinase (Fig. 5)[38–42].

Our data lead us to propose an unusual mechanism, whereby the dimerization of PK-1 via an atypical dimer interface facilitates constitutive phosphorylation of its substrates. This dimerization mode stabilizes elements typically flexible in kinases, the Gly-rich and activation loops, and by locking the kinase into a rigid, active conformation marked by an intact R-spine and β3-strand K49:αC helix E60 salt bridge, predisposes PK-1 to binding ATP in a productive orientation for catalysis. The β3-strand residue, K49, which is known to mediate ATP α- and β-phosphate binding in

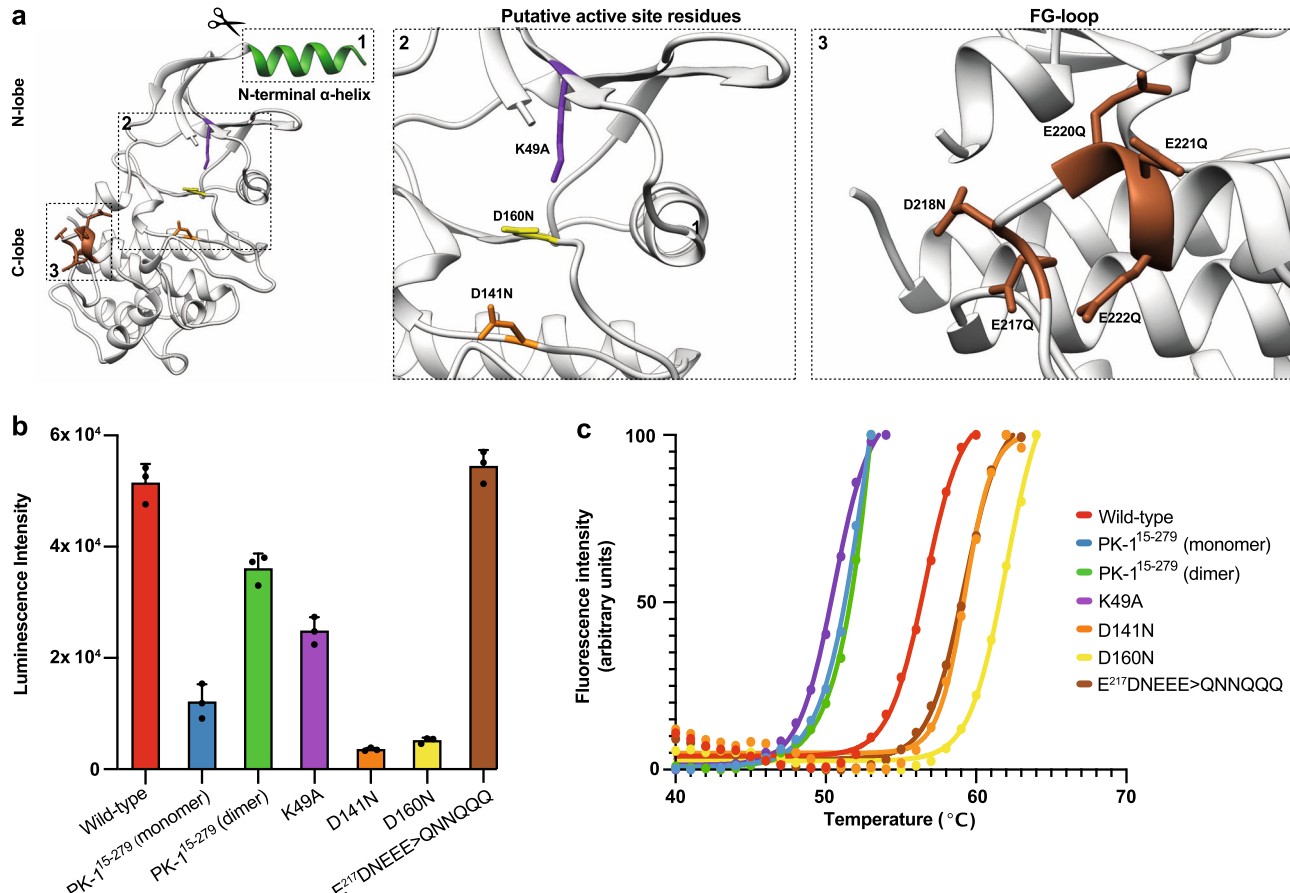

**Fig. 4 Biochemical characterisation of PK-1. a** Residues mutated in the present study are shown as sticks on the PK-1 crystal structure solved in this study (Fig. 1), including the truncated N-terminal α-helix (green ribbon) to generate PK-1[15-279] (box 1) and zoomed views of the putative active site residues (box 2) and the patch of acidic residues (brown sticks) within the αF-αG loop (box 3) shown on the right. **b** In vitro kinase activities of the PK-1 wild-type and mutant constructs. Individual data points are plotted; the bar and error bars shown represent mean ± SD of three independent ADP-Glo assays. **c** Thermal melt curves of the PK-1 wild-type and mutant constructs, performed using differential scanning fluorimetry confirms that these variants are folded. Data represent mean of two independent experiments. Data are plotted throughout for wild-type PK-1 in red, and mutants are color-coded: truncated PK-1 monomer in blue and dimer in green; K49A, purple; D141N, orange; D160N, yellow; and E[217]DNEEE>QNNQQQ, brown.

conventional protein kinases and pseudokinases[35,43–46], and serve an indispensable role in catalysis in conventional kinases[47,48], could be substituted to Ala and retain ~50% of wild-type activity. This contrasts with Asn substitution of the canonical catalytic Asp (D160) and Mg²⁺-binding Asp (D141), which lead to almost complete ablation of catalytic activity. In addition to the dispensability of K49 for catalytic activity, the Gly-rich loop (D[24]ESdteSYS) is divergent relative to those in conventional protein kinases (GδGXφG). We propose that the structural rigidification in PK-1 dimers that arises from the parallel side-to-side dimerization mode supersedes the necessity of Gly-rich loop and activation loop flexibility and the β3-strand Lys for ATP-binding and catalysis. Although this dimerization mode has not been reported for other kinases or pseudokinases[49], our findings raise the prospect more broadly that inter-kinase/pseudokinase interactions may serve functions to rigidify the core active site elements responsible for ATP-binding and positioning for catalysis.

PK-1 contains an acidic insertion within the αF-αG loop (E[217]DNEEE), which based on our bioinformatic analysis, is present in *Baculoviridae* PK-1 sequences. Considering PK-1's function as a kinase that targets Arg-rich substrates, we hypothesized that this sequence could serve a function in recruitment of polybasic substrates. However, in our hands, mutation of this sequence to neutralize charge did not impact catalytic activity relative to wild-type protein. As a result, our data suggest that, at

least under conditions where substrate is in excess, this element, which is highly mobile in molecular dynamics simulations, does not markedly influence substrate recognition. One possibility is that the acidic sequence of the atypical Gly-rich loop on the opposing lip of the active site may compensate for loss of the acidic αF-αG loop sequence to facilitate recognition of basic substrates. Functionally, recognition of the Arg-rich substrate, p6.9, by PK-1 occurs in nuclear occlusions where the substrate is highly concentrated and, accordingly, would not be expected to rely on the complementary charges of p6.9 and PK-1's acidic αF–αG loop for phosphorylation. Instead, it appears that the role of PK-1 phosphorylation is to reduce the net positive charge of p6.9 to disrupt DNA interaction so transcription of viral very late genes can proceed. In support of this idea, 7 serines within p6.9 were identified as crucial PK-1 substrates, where their mutation to Ala prevented progression of very late gene transcription[17]. Deletion of PK-1 from *Autographa californica* multiple nucleopolyhedrovirus compromised p6.9 phosphorylation in Sf9 insect cells, revealing an important role for PK-1 in promoting hyperphosphorylation of Ser and Thr residues and apparently priming p6.9 for Arg phosphorylation. Because some Arg and Ser/Thr phosphorylation was observed within p6.9 in the absence of PK-1[17], this suggests some redundancy among the p6.9-targeting kinases encoded within baculoviral genomes. Furthermore, these data indicate that some viral kinases catalyse Arg phosphorylation

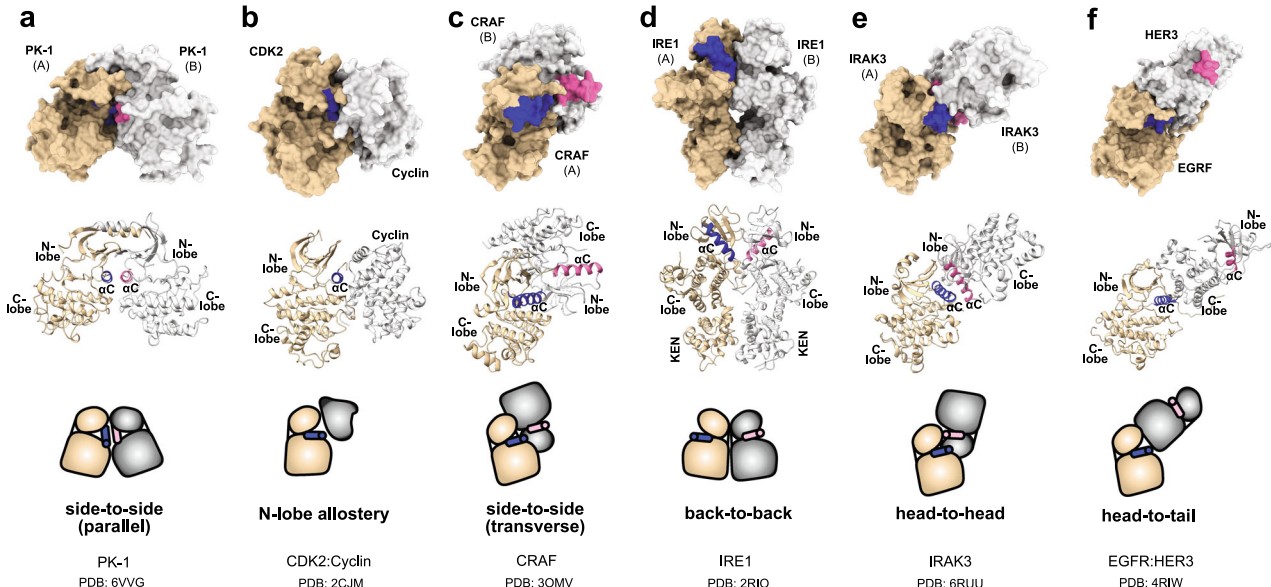

**Fig. 5 Comparison of known modes of kinase dimerization and allostery. a** Parallel side-to-side homodimer PK-1 (PDB 6VVZ; solved in this study); **b** allosteric regulation of CDK2 via N-lobe binding to Cyclin (PDB 2CJM)[77]; **c** transverse side-to-side homodimer CRAF (PDB 3OMV)[38]; **d** back-to-back homodimer IRE1 (PDB 2RIO)[41]; **e** head-to-head homodimer IRAK3 (PDB 6RUU)[40]; **f** head-to-tail heterodimer EGFR:HER3 (PDB 4RIW)[42]. Each allosteric or dimerization mode is illustrated by molecular surface (top panel), cartoon ribbon (middle panel) and schematic representation (lower panel). Protomer A and B are coloured tan and grey, respectively. The αC-helix (αC) from protomer A and B are highlighted in dark blue and pink, respectively. The N-lobe (N) and C-lobe (C) are shown, and the KEN domain that contributes to the IRE1 dimer interface is indicated in **d**.

that is primed by Ser/Thr phosphosites within p6.9. Thus, whether the Gly-rich loop and αF-αG loop acidic sites work in concert to contribute to Arg-rich peptide substrate recognition remains of outstanding interest.

The dimerization mode observed for PK-1 in our structural and biophysical studies is unusual among protein kinases. The combination of allostery, where each protomer locks its dimer-opposed partner into a rigid conformation primed for catalysis, and the parallel dimerization mode with one subunit facing in each direction, distinguish PK-1 from other reported modes of kinase dimerization. Such a dimerization mode would allow processive phosphorylation of repetitive substrates, like protamines and p6.9 in the case of PK-1, via two active sites poised in a catalytically-active conformation by virtue of the locked, side-to-side dimerization. Knowledge of the extent to which side-to-side dimerization is used more broadly in nature among viral and eukaryotic protein kinases to regulate catalytic activity awaits future detailed studies.

## Methods

**Protein expression and purification**. The gene coding for full-length *C. pomonella* GV PK-1 (isolate Mexico/1963; Uniprot Q91F41) was codon-optimised and synthesized for *E. coli* expression (Genewiz; Supplementary Table 1) and sub-cloned into the expression vector pPROEX Htb as an in-frame fusion with a TEV protease-cleavable N-terminal hexahistidine tag. The D141N and D160N mutations were introduced into the wild-type template using oligonucleotide-directed overlap PCR (Supplementary Table 1), with remaining mutant sequences synthesized by GeneWiz. All insert sequences were verified by Sanger sequencing (AGRF, Australia). Wild-type and mutant PK-1 constructs were expressed in *E. coli* BL21-CodonPlus-RIL (Agilent) or C41 (DE3) cells cultured in Super Broth supplemented with ampicillin (100 μg mL$^{-1}$) at 37 °C with shaking at 220 rpm to an OD$_{600}$ of ~0.6–0.8. Protein expression was induced by the addition of isopropyl β-D-1-thiogalactopyranoside (250 μM) and the temperature was lowered to 18 °C for incubation overnight. Cell pellets were resuspended in lysis buffer (20 mM HEPES, pH 7.5, 200 mM NaCl, 5% glycerol, 0.5 mM TCEP), supplemented with Complete protease cocktail inhibitor (Roche), and lysed by sonication. Cell debris and insoluble material was pelleted via centrifugation at 45,000 × *g* and the lysate was incubated with pre-equilibrated Ni-NTA agarose (HisTag, Roche) at 4 °C for 1 h with gentle agitation. Ni-NTA beads were then pelleted via centrifugation and washed thoroughly with wash buffer (20 mM HEPES, pH 7.5, 200 mM NaCl, 5 mM imidazole, 5% glycerol, 0.5 mM TCEP). Bound protein was eluted from the beads

using elution buffer (20 mM HEPES, pH 7.5, 200 mM NaCl, 250 mM imidazole, 5% glycerol, 0.5 mM TCEP), filtered through a 0.45-μm filter, mixed with 300 μg of recombinant His$_6$-TEV and dialysed overnight in size exclusion buffer (20 mM HEPES, pH 7.5, 200 mM NaCl, 5% glycerol) at 4 °C. Following protease cleavage, the dialysate was incubated with Ni-NTA agarose (Roche) pre-equilibrated in the same buffer at 4 °C for 1 h to remove TEV and uncleaved protein. Following incubation, the sample was centrifuged to eliminate particulates, the supernatant 0.45-μm filtered, concentrated via centrifugal ultrafiltration (30 kDa molecular weight cut-off; Millipore) and loaded onto a HiLoad 16/600 Superdex 200 prep grade size exclusion column (Cytiva) equilibrated in size exclusion buffer. For AUC experiments, glycerol was omitted from the buffer. The final purity of the eluted protein was assessed by SDS-PAGE (Supplementary Fig. 1). Protein that was not immediately used in experiments was aliquoted, flash-frozen in liquid nitrogen and stored at −80 °C.

**Crystallization, data collection and structure determination**. For crystallization trials, the gene encoding full-length *C. pomonella* GV PK-1 was amplified from genomic DNA (using primer sequences reported in Supplementary Table 1) and subcloned into the Gateway expression vector, pDEST566. The resulting protein, bearing a TEV protease-cleavable N-terminal hexahistidine and maltose-binding protein tag, was expressed in *E. coli* Rosetta (DE3) cells cultured in Luria Broth containing ampicillin (100 μg mL$^{-1}$) at 37 °C with shaking at 220 rpm to an OD$_{600}$ of ~0.6–0.8. Protein expression was induced by the addition of isopropyl β-D-1-thiogalactopyranoside (100 μM) and the temperature was lowered to 18 °C for incubation overnight. Purification was performed essentially as described above, but Ni$^{2+}$-chromatography was performed using a 5 mL HisTrap FF NiNTA column (Cytiva). Protein eluted from a HiLoad 16/600 Superdex 200 size exclusion column (Cytiva) in 50 mM HEPES, pH 8.0, 150 mM NaCl, 2 mM TCEP was concentrated to 5 mg mL$^{-1}$ (152 μM) and subjected to sitting-drop vapour diffusion experiments at 18 °C using the JCSG + and PACT premier (Molecular Dimensions) commercial crystallization screens. A volume of 0.15 μL protein solution and 0.15 μL reservoir solution was equilibrated over 50 μL of reservoir solution using an Oryx 4 Protein Crystallisation Robot (Douglas Instruments LTD, Berkshire, UK). Spontaneously nucleating PK-1 crystals were of insufficient size for data collection, so a fine screen was developed from initial hits. Because initial hits did not diffract beyond 6 Å, we supplemented the protein solution with 10 mM MnCl$_2$ and 1 mM adenosine monophosphate (AMP) and obtained suitable PK-1 crystals in 300 mM NH$_4$Cl and 16.5% (w/v) PEG 3350 at 18 °C. For data collection, the crystals were cryoprotected in this condition supplemented with 20% (w/v) glycerol and then flash-frozen. Data were processed using XDS[50] and AIMLESS from the CCP4 program suite[51] to 2.01 Å in the space group *P* 2$_1$2$_1$2$_1$. The structure of PK-1 was determined by molecular replacement in PHASER[52] using the human Ca$^{2+}$/Calmodulin-dependent protein kinase II gamma (CaMK2G, PDB accession: 2V7O)[53]. The resulting model was improved by iterative rounds of refinement using PHENIX[54] and manual building in COOT[55], which included the

addition of AMP and water molecules. Due to insufficient electron density, residues 1, 2, 26–29 and 217–222 in chain A, and residues 1, 2, 27–31, 105–111 and 279 in chain B were not modelled. Final model validation was performed using MOL-PROBITY[56]. The dimer interface was analysed using PDBePISA[57]. All structural graphics were prepared using UCSF Chimera[58] and PyMOL. All data collection and refinement statistics are summarized in Supplementary Table 2.

**Sequence and phylogenetic analysis**. NCBI BLAST was performed using full length $Cp$GV PK-1 sequence as the query searched against the non-redundant protein sequences (NR) database. The kinase domains of the sequences were aligned using MAPGAPS[59] and a curated profile built using structural alignments of eukaryotic protein kinases (ePKs) and the $Cp$GV PK-1 crystal structure (PDB: 6VVG) and purged to 90% identity. A maximum-likelihood tree was constructed using FaSTtree 2.1.11 (ref. [60]) and visualized using Interactive Tree Of Life (ITOL)[61]. Evolutionary constraints imposed on PK-1 sequences were quantified using Contrast Hierarchical Alignment Interaction Network (CHAIN)[62] analysis, which quantifies the extent to which aligned residues in the foreground alignment (PK-1 sequence alignment) diverge from the corresponding position in the background alignment (alignment of representative ePK sequences).

**Analytical ultracentrifugation**. Sedimentation velocity experiments were performed in a Beckman Coulter Optima analytical ultracentrifuge using double sector epon-charcoal centerpieces fitted with sapphire windows in an An-50 Ti eight-hole rotor at 20 °C. Data were obtained at 50,000 rpm using the absorbance optical system at 280 nm, measuring protein at 0.9 mg mL$^{-1}$ (27 μM) against a buffer reference (20 mM HEPES, pH 7.5, 200 mM NaCl). Sedimentation data were fitted to a continuous size distribution $[c(s)]$ model and converted to a standardized sedimentation coefficient ($s_{20,w}$) distribution using SEDFIT[63]. The buffer density, buffer viscosity and an estimate of the partial specific volume of the protein sample, based on the amino acid sequence were calculated using SEDNTERP. All data collection and analysis statistics are summarized in Supplementary Table 3.

**Small-angle X-ray scattering**. SAXS data were collected at the Australian Synchrotron SAXS/WAXS beamline using an inline co-flow size-exclusion chromatography setup to minimize sample dilution and maximize signal-to-noise[64]. Protein at 5 mg mL$^{-1}$ (152 μM) was injected (80 μL) onto an inline Superdex 200 5/150 Increase GL column (Cytiva), equilibrated with size exclusion buffer (20 mM HEPES, pH 7.5, 200 mM NaCl, 5% glycerol) at 12 °C, at a flow rate of 0.2 mL min$^{-1}$ using an established protocol[65,66]. 2D intensity plots were radially averaged, normalized against sample transmission, and background-subtracted using the Scatterbrain software package (Australian Synchrotron). The ATSAS software package was used to perform the Guinier analysis (PrimusQT)[67], to calculate the pairwise distribution function $P(r)$ and the maximum interparticle dimension ($D_{max}$), and to evaluate the solution scattering against the crystal structure solved in this study (CRYSOL)[68]. The molecular mass of each sample was estimated using the SAXS-MoW2 package[69]. All data collection and processing statistics are summarized in Supplementary Table 4.

**Analytical size exclusion chromatography**. Analytical size exclusion chromatography runs were performed in size exclusion buffer (20 mM HEPES, pH 7.5, 200 mM NaCl, 5% glycerol) on a Superdex 200 5/150 Increase GL column (Cytiva). Purified protein was injected (100 μL) onto the column at concentrations varied from 1 to 5 mg mL$^{-1}$ (30–152 μM). Gel filtration standard (BioRad, CA) was used to define the molecular weight marker elution volumes (1.35-670 kDa).

**Molecular dynamic simulations**. Unbiased full atom MD simulation of apo $Cp$GV PK1 in the monomeric and dimeric forms were performed using GROMACS 2018.1. Hetero atoms and water molecules were removed from the co-crystal structure of $Cp$GV PK-1 dimer bound to AMP molecules (PDB ID: 6VVG). The main and sidechain atoms for residues 27–29 in chain A and residues 27–79, 105–110 in chain B were modelled using RosettaLoop[70] using the cyclic coordinate descent (CCD) protocol. Chain A was used for the monomer simulation whereas both chains were used for the dimer simulation. The hydrogen atoms were converted to virtual sites to remove the fastest vibrational freedom. The protein was parameterized with Amber99SB-ILDN and solvated with TIP3P waters modelled in a dodecahedron box. The charges were neutralized by addition of sodium and chloride ions. The neighbour list for nonbonded interactions was defined using Verlet cutoff[71]. Particle Mesh Ewald (PME) was used to calculate long-range interactions. Energy minimization was performed with steepest-descent algorithm followed by conjugate descent with Fmax less than 500kJmol$^{-1}$nm$^{-1}$. The canonical ensemble was carried out by heating the system from 0 K to 310 K, using velocity rescaling for 100 ps[72]. The isothermal–isobaric ensemble (P = 1 bar, T = 310 K) was carried out using the Berendsen barostat for 100 ps[73]. The unrestrained MD productions were collected using a time step of 5 fs. The trajectories were processed and analysed using the built-in GROMACS tools. Visualization was done using PyMOL 2.3.2.

**In vitro kinase activity assay**. The in vitro kinase activity assay was performed with an ADP-Glo kinase assay kit (Promega), following standard procedures. Briefly, a 25 μL kinase reaction contained 0.1 mg mL$^{-1}$ PK-1, 0.5 mg mL$^{-1}$ p6.9 peptide (with the sequence MVRRRR SRSPNR RRSYRS RSRSRS RSRSRS RSRSRS PYRS, synthesized to >95% purity by Mimotopes Pty Ltd), and 1 mM ATP in kinase reaction buffer (20 mM HEPES, pH 7.5, 20 mM MgCl$_2$, 2 mM MnCl$_2$, 1 mM DTT, and 0.1 mg mL$^{-1}$ BSA). Each reaction was performed at 25 °C for 1 h and terminated by the addition of the ADP-Glo reagent. After incubation for 40 min with the ADP-Glo reagent, the Kinase Detection reagent was added and after an incubation time of 1 h, the luminescence was detected with a microplate reader (CLARIOstar, BMG LabTech).

**Thermal shift assays**. Thermal shift assays were performed as described previously[44,74] using a Corbett Real Time PCR machine with proteins diluted in 20 mM HEPES, pH 7.5, 200 mM NaCl to 10 μg in a total reaction volume of 25 μL. SYPRO Orange (Molecular Probes, CA) was used as a probe with fluorescence detected at 530 nm. Two independent assays were performed for wild-type and mutant PK-1 constructs; averaged data are shown for each in Fig. 4.

**Reporting summary**. Further information on research design is available in the Nature Research Reporting Summary linked to this article.

## Data availability
Data are available from the corresponding author upon reasonable request. The atomic coordinates for PK-1 have been deposited in the Protein Data Bank with the accession code 6VVG. Source data are provided with this paper.

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

## Acknowledgements

We thank staff at the Australian Synchrotron MX2 and SAXS/WAXS beamlines for their assistance in data collection and the New Zealand Synchrotron Group for enabling access; Yee-Foong Mok (Bio21 Institute, University of Melbourne) for assistance with analytical ultracentrifugation experiments. J.M.M. gratefully acknowledges National Health and Medical Research Council of Australia support (Investigator grant 1172929 and IRIISS grant 9000653) and the Victorian Government Operational Infrastructure Support Scheme. Funding for N.K. from the National Institutes of Health (R01GM114409) is acknowledged. P.M. gratefully acknowledges support of the New Zealand Royal Society Marsden (grant UOA221).

## Author contributions

M.R.O. and C.R.H. designed and performed experiments, analysed data and co-wrote the paper with J.M.M.; J.R.K. and D.C.G. contributed to protein crystallization, X-ray data collection and structure determination; S.S. and N.K. performed and analysed molecular dynamic simulations, and conducted phylogenetic analyses; L.Y.L., S.N.Y., J.J.S., A.I.W. and I.S.L. performed experiments and analysed data; P.M. and J.M.M. supervised the project and contributed to experimental design and data analysis. All authors commented on the manuscript.

## Competing interests

The authors declare no competing interests.
