## [Peer Review File · Nature Communications]

REVIEWERS' COMMENTS

Reviewer #1 (Remarks to the Author):

The manuscript of Oliver et al "Granulovirus PK-1 kinase activity relies on a side-to-side dimerization mode centered on the regulatory α C helix" is a solid and thorough work presenting a novel structure of viral protein kinase PK-1. It will be very useful for a wide circle of scientists studying protein kinase and definitely can be recommended for publication in Nature Communications. My only concern is that the discovered mechanism of activation is taken out of context. Activation of kinases via dimerization is a well-known phenomenon and needs to be described at least briefly for potential readers. It is usually recommended to avoid any sensationalistic language like "unprecedented" (pp. 4,12) as it is unclear to what extent this is an unprecedented result. There is nothing new about protein kinase activation via dimerization: two types of dimers should be mentioned – symmetric (BRAF) and asymmetric (EGFR). Activation of kinases by stabilization of their C-helices is also very well known, e.g. PDK1 via small molecule binding in the PIF pocket or cyclin induced activation of CDKs. The new reported PK-1 dimer is also a symmetric dimer, similar to BRAF, however the binding interface is not at the α C-b4 region but at the C-helix (similar to CDKs). Indeed, this kind of dimerization was never observed and is of great interest.

A few minor notes:

the word "centered" in the title is misspelled;

two beta strands in the C-lobe are labeled as b6 and b7. There is an established nomenclature for conserved secondary structures in protein kinases and, according to this nomenclature, these strands are b7 and b8. For some reason the b6-b9 beta sheet at the N-terminus of the Catalytic loop is missing although the Activation loop looks like in a classic extended conformation. I would encourage the authors to double check their secondary structure assignments.

Reviewer #2 (Remarks to the Author):

The manuscript titled "Granulovirus PK-1 kinase activity relies on a side-to-side dimerization mode centered on the regulatory α C helix" reports the crystal structure of *Cydia pomonella* granulovirus PK-1 which occurs as a rigid dimer. The manuscript provides a detailed study regarding the arrangement of dimers and behavior of intact dimer conformation in solutions. The authors made an excellent study showing the behavior of catalytically active site conformation and the amino acid residues lining the active site. Additional validation studies were also carried out through molecular dynamic simulations. The deposited crystal structure of the dimer in protein data bank will be an invaluable contribution to the scientific community working in this research area. It is original, nice flow and is well organized. I believe the manuscript is timely and will get the attention of many researchers in the short and long runs. The manuscript is publishable in the present form with the following comments:

Comments:

1. It would be nicer to see; is there any allosteric effect between the dimers at the interface, especially, at the catalytic site? (This is not a hindering point for publication).
2. Figure 1 is not clearly visible.
3. In Figure 3 only residue numbers versus RMSF graph were given, however, radius of gyration (shows compactness of the enzyme-ligand complex during simulation time) versus time and RMSD graph of equilibrated enzyme dimer versus time would also be more informative in assessing the dynamic behavior of the enzyme dimer.

Reviewer #3 (Remarks to the Author):

This manuscript shows the molecular mechanism of granulovirus PK-1 kinase activity which relies on a side-to-side dimer configuration. Structural and biophysical experiments depicted this unprecedented dimerization. As one of them, the unique helix in the N-terminal contributes to the dimerization and exertion of activity. The results broaden the scope of studies about kinase regulatory mechanisms. So, the manuscript should be accepted after the following the minor revision.

1) Authors should insert the sentences on interaction detail between the α C helices in the dimer. In abstract, authors stated the importance of this interaction for dimerization but there is little statement about it in the result section.

REVIEWERS' COMMENTS

Reviewer #1 (Remarks to the Author):

The manuscript of Oliver et al “Granulovirus PK-1 kinase activity relies on a side-to-side dimerization mode centered on the regulatory α C helix” is a solid and thorough work presenting a novel structure of viral protein kinase PK-1. It will be very useful for a wide circle of scientists studying protein kinase and definitely can be recommended for publication in Nature Communications. My only concern is that the discovered mechanism of activation is taken out of context. Activation of kinases via dimerization is a well-known phenomenon and needs to be described at least briefly for potential readers. It is usually recommended to avoid any sensationalistic language like “unprecedented” (pp. 4,12) as it is unclear to what extent this is an unprecedented result. There is nothing new about protein kinase activation via dimerization: two types of dimers should be mentioned – symmetric (BRAF) and asymmetric (EGFR). Activation of kinases by stabilization of their C-helices is also very well known, e.g. PDK1 via small molecule binding in the PIF pocket or cyclin induced activation of CDKs. The new reported PK-1 dimer is also a symmetric dimer, similar to BRAF, however the binding interface is not at the α C-b4 region but at the C-helix (similar to CDKs). Indeed, this kind of dimerization was never observed and is of great interest.

We thank the reviewer for their positivity and for prompting further discussion on the important point that the mode of dimerization observed in the PK-1 structure is distinct from those previously reported. We have now removed any claims of priority (such as “unprecedented”), as requested by the reviewer, and also in keeping with the journal guidelines, and instead refer to the conformation as “unusual”. To illustrate precisely how unusual, in keeping with the reviewer’s suggestion, we have included an additional figure that summarizes the diverse modes of kinase dimerization reported previously and the mentioned allosteric mode of Cyclin binding to CDKs. This is now presented as **Figure 5** in the revised version, with additional Discussion now included on lines 303-311.

*A few minor notes:
the word “centered” in the title is misspelled;*

We thank the reviewer for noticing this. We had somewhat of an existential crisis in preparing the work for publication and were torn between UK and US spelling. We have now revised to “centered” for consistency with style throughout the rest of the manuscript.

two beta strands in the C-lobe are labeled as b6 and b7. There is an established nomenclature for conserved secondary structures in protein kinases and, according to this nomenclature, these strands are b7 and b8. For some reason the b6-b9 beta sheet at the N-terminus of the Catalytic loop is missing although the Activation loop looks like in a classic extended conformation. I would encourage the authors to double check their secondary structure assignments.

We greatly appreciate the reviewer picking up this error. We have now corrected, with enormous thanks, in both the figure and the results.

Reviewer #2 (Remarks to the Author):

*The manuscript titled “Granulovirus PK-1 kinase activity relies on a side-to-side dimerization mode centered on the regulatory α C helix” reports the crystal structure of *Cydia pomonella* granulovirus PK-1 which occurs as a rigid dimer. The manuscript provides a detailed study regarding the arrangement of dimers and behavior of intact dimer conformation in solutions. The authors made an excellent study showing the behavior of catalytically active site conformation and the amino acid residues lining the active site. Additional validation studies were also carried out through molecular dynamic simulations. The deposited crystal structure of the dimer in protein data bank will be an invaluable contribution to the scientific community working in this research area.*

It is original, nice flow and is well organized. I believe the manuscript is timely and will get the attention of many researchers in the short and long runs.

We thank the reviewer for their kind appraisal of the work and their support for publishing this study.

The manuscript is publishable in the present form with the following comments:

Comments:

1. It would be nicer to see; is there any allosteric effect between the dimers at the interface, especially, at the catalytic site? (This is not a hindering point for publication).

This is an important point that we have taken the opportunity to expand on in revision. We have now included additional panels to highlight the allosteric interactions mediated within the dimers, and we have added a further panel (revised Figure 1e) to illustrate the contacts between α C helices (also suggested by Reviewer 3) and have expanded Figure 1d to illustrate the role of the dimer partner α C in allostery. We have added further comment on this point (lines 113-121).

2. Figure 1 is not clearly visible.

Our apologies for the clarity of this figure. We sought to reduce the size of the figure for review purposes, however this has compromised the quality. We have attended to this during revision and expect it is now more suitable for publication.

3. In Figure 3 only residue numbers versus RMSF graph were given, however, radius of gyration (shows compactness of the enzyme-ligand complex during simulation time) versus time and RMSD graph of equilibrated enzyme dimer versus time would also be more informative in assessing the dynamic behavior of the enzyme dimer.

We agree and have now included an additional figure to present these data as Supplementary Figure 4 and some accompanying description on lines 224-226. We thank the reviewer for this suggestion.

Reviewer #3 (Remarks to the Author):

This manuscript shows the molecular mechanism of granulovirus PK-1 kinase activity which relies on a side-to-side dimer configuration. Structural and biophysical experiments depicted this unprecedented dimerization. As one of them, the unique helix in the N-terminal contributes to the dimerization and exertion of activity. The results broaden the scope of studies about kinase regulatory mechanisms. So, the manuscript should be accepted after the following the minor revision.

We thank the reviewer for their kind evaluation of this work.

1) Authors should insert the sentences on interaction detail between the α C helices in the dimer. In abstract, authors stated the importance of this interaction for dimerization but there is little statement about it in the result section.

We are grateful for this suggestion and have now included additional text (also in line with reviewer 2's suggestion) as lines 113-121 on page 5, and an additional panel (Figure 1e), to more clearly show the contacts between α C helices.